# 6mA-DNA-binding factor Jumu controls maternal-to-zygotic transition upstream of Zelda

Shunmin He [1,2,8], Guoqiang Zhang [1,8], Jiajia Wang[2,8], Yajie Gao[1,3,8], Ruidi Sun[1,4], Zhijie Cao[1], Zhenping Chen[1], Xiudeng Zheng [5], Jiao Yuan[2], Yuewan Luo[1,3], Xiaona Wang[1], Wenxin Zhang[1,4], Peng Zhang[2], Yi Zhao[6], Chuan He [7], Yi Tao [5], Qinmiao Sun[1,3] & Dahua Chen[1,3]

A long-standing question in the field of embryogenesis is how the zygotic genome is precisely activated by maternal factors, allowing normal early embryonic development. We have previously shown that N6-methyladenine (6mA) DNA modification is highly dynamic in early *Drosophila* embryos and forms an epigenetic mark. However, little is known about how 6mA-formed epigenetic information is decoded. Here we report that the Fox-family protein Jumu binds 6mA-marked DNA and acts as a maternal factor to regulate the maternal-to-zygotic transition. We find that *zelda* encoding the pioneer factor Zelda is marked by 6mA. Our genetic assays suggest that Jumu controls the proper zygotic genome activation (ZGA) in early embryos, at least in part, by regulating *zelda* expression. Thus, our findings not only support that the 6mA-formed epigenetic marks can be read by specific transcription factors, but also uncover a mechanism by which the Jumu regulates ZGA partially through Zelda in early embryos.

[1] State Key Laboratory of Membrane Biology, Institute of Zoology, Chinese Academy of Sciences, Beijing 100101, China. [2] Key Laboratory of RNA Biology, Center for Big Data Research in Health, Institute of Biophysics, Chinese Academy of Sciences, Beijing 100101, China. [3] School of Life Sciences, University of Chinese Academy of Sciences, Beijing 100049, China. [4] School of Life Sciences, Anhui Agricultural University, Hefei, Anhui 230036, China. [5] Centre for Computational and Evolutionary Biology, Key Laboratory of Animal Ecology and Conservational Biology, Institute of Zoology, Chinese Academy of Sciences, Beijing 100101, China. [6] Advanced Computer Research Center, Institute of Computing Technology, Chinese Academy of Sciences, Beijing 100190, China. [7] Department of Chemistry and Institute for Biophysical Dynamics, Howard Hughes Medical Institute, The University of Chicago, Chicago, IL 60637, USA. [8] These authors contributed equally: Shunmin He, Guoqiang Zhang, Jiajia Wang, Yajie Gao. Correspondence and requests for materials should be addressed to Y.T. (email: yitao@ioz.ac.cn) or to Q.S. (email: qinmiaosun@ioz.ac.cn) or to D.C. (email: chendh@ioz.ac.cn)

Methylated bases, such as 5-methylcytosine (5mC) and N6-methyladenine (6mA), are the most abundant types of DNA modification in the genomes of diverse species[1,2]. 5mC modification has been studied extensively in higher eukaryotes. Previous studies have suggested that the dynamic regulation of 5mC plays important roles in regulating chromatin architecture and gene expression during mammalian development and adult tissue homeostasis[3–6]. Contrary to the significant progresses made in understanding cytosine modification, the role of 6mA was considered less important in higher eukaryotes[7]. Recent studies suggest that 6mA associates with gene expression in multiple eukaryotic species, including algae, worm, fly, and mammals, indicating that 6mA has potential epigenetic roles in eukaryotes[8–13]. However, little is known about mechanisms of how 6mA-related epigenetic codes are formed and interpreted to direct gene expression programs.

At the early embryonic stages, all animal fertilized embryos undergo a critical process, namely the maternal-to-zygotic transition (MZT). During MZT, a proportion of the oocyte-loaded maternal products are cleared and the zygotic genome is activated. While maternal mRNA clearance is triggered by a number of RNA-binding proteins, zygotic genome activation (ZGA) is primarily instructed by a small number of pioneer transcription factors[14–17]. In *Drosophila*, early embryogenesis is characterized by a series of rapid synchronous nuclear divisions. The zygotic genome is almost quiescent in the first few rounds of nuclear division (before cycles 7 or 8)[16,18], and becomes widely activated when the MZT ends at the 14th cycle[19]. It has been shown that Zygotic early *Drosophila* activator (Zelda) (or called Vfl) functions as a pioneer transcription factor and accesses the early embryonic genome by binding sequence-specific motifs (referred to as TAGteam sites), and subsequently increases chromatin accessibility for other transcription factors, thus ensuring coordinated gene expression during MZT[20–24]. Of note, the proper expression of Zelda is critical for early embryogenesis, since either loss of or overexpression of Zelda leads to defects of embryonic development[20–22,24], raising a possibility that an uncharacterized mechanism coordinates with the pioneer factor Zelda to regulate early embryonic events.

We have previously shown that 6mA is highly dynamic in *Drosophila* early embryos. Notably, the timing window of the 6mA dynamics almost corresponded to the MZT process during early embryogenesis[17,23,25]. We speculate that 6mA may contribute to MZT by forming an epigenetic code that can be recognized by maternal factors in early embryos. In this study, we show that the Fox family protein Jumu functions as a maternal transcription factor and regulates embryonic gene expression by preferentially binding the 6mA-marked DNA. Importantly, we find that *zelda* is marked with 6mA and regulated by Jumu. Our genetic analyses show that partial knockdown of Zelda significantly suppresses the embryonic lethal phenotype induced by loss of maternal Jumu. Together, our findings suggest that Jumu preferentially binds 6mA-marked DNA and controls MZT, at least in part through regulating Zelda.

## Results

### Landscape of 6mA modification in early embryonic genomes.
To explore the potential role of 6mA in MZT, we first characterized the genome-wide features of 6mA. We collected genomic DNA from 0.75-h (nearly pre-MZT and pre-ZGA), 3-h (post-ZGA), and 6-h (post-MZT) post-fertilization embryos (see Methods; Fig. 1a) and then employed an anti-6mA antibody (Abcam) to perform DNA immunoprecipitation (DNA-IP) experiments (see Methods; Supplementary Figure 1a–c). The IPed

DNAs were then subjected to the paired-end (125 bp) high-throughput sequencing (Supplementary Figure 1d).

We identified 17,528 6mA peaks at the 0.75-h stage and several thousands of peaks at the 3-h and 6-h stages (Fig. 1b and Supplementary Figure 1e, f). The signal strength of the 6mA peaks in the 0.75-h sample was much stronger than that in the two later stages (Fig. 1c and Supplementary Figure 1g). Moreover, about 80% (13,897/17,528) of 6mA peaks identified at the 0.75-h stage had disappeared at the 3-h stage, although these peak regions still retained faint 6mA signals in the 3-h samples (Fig. 1b, d and Supplementary Figure 1e). Additionally, about 20% of peaks identified at the 0.75-h stage were also present at the 3-h stage, but 6mA signals in these common peaks were weaker at the 3-h stage (Fig. 1d), suggesting that 6mA modification were dynamically regulated during early embryogenesis (Fig. 1e).

We have previously shown that DMAD, a 6mA demethylase in *Drosophila*, is essential for 6mA demethylation and embryogenesis. To investigate whether DMAD is involved in regulating the 6mA landscape in early embryo genomes, we collected genomic DNA from *DMAD* knockdown embryos (Supplementary Figure 1h) at the stage of 6-h post fertilization, and performed DNA-IP followed by paired-end high-throughput sequencing. In contrast to the control (6-h wild-type embryos), from which we obtained 2,447 6mA peaks, the number of 6mA peaks from *DMAD* knockdown embryos was increased to 8,367 (Fig. 1f and Supplementary Figure 1i). These peaks were highly overlapped with those in 0.75-, 3- and 6-h wild-type samples (Fig. 1f). Moreover, when compared with the 6-h wild-type samples, approximately 79% of the gained 6mA peaks in 6-h *DMAD* knockdown embryos were also found in the 0.75-h or 3-h samples (Fig. 1f and Supplementary Figure 1i) and the signal of these gained 6mA peaks also exhibited a dynamic pattern from 0.75-h to 6-h (Fig. 1g, h), suggesting that DMAD plays an important role in controlling dynamics of 6mA modification in early embryonic genomes.

### 6mA can mark coding genes in early embryos.
We have previously reported that 6mA marks transposon sequences in ovaries. Consistent with this observation, we found that, in addition to transposons, such as LTR and LINE, a portion of 6mA peaks in early embryo was enriched with simple repeat regions (Supplementary Figure 2a, b). As shown in Fig. 2a, about 33~43% of 6mA peaks were mapped to the intronic regions, and 36~45% of 6mA peaks were found to locate in intergenic regions at different embryonic stages. Further analyses revealed that 49–72% of intronic 6mA peaks and 63–74% of intergenic 6mA peaks were located in transposons and simple repeat regions (Fig. 2a). Of note, although a large proportion of 6mA peaks were mapped to the simple repeat and transposons, many of these peaks were also found to locate nearby coding genes or within the introns, and were considered as coding gene-related 6mA peaks in this study. In total, we identified 54–62% of coding gene-related 6mA peaks (see Methods; Fig. 2a).

To test whether 6mA modification is associated with the coding gene expression in embryos, we performed RNA-seq with embryo samples at 0.75-, 3-, and 6-h stages and found 6,723 differentially expressed genes between at least two different stages. These genes were then clustered into several groups (Supplementary Figure 2c). Based on their expression level trends, we defined group 1 (489 genes) as a typical group of zygotically activated genes (Supplementary Figure 2c, d), and found that 6mA-marked genes were significantly enriched ($P = 5.34e-28$) in zygotically activated genes (Fig. 2b).

We next tested whether 6mA influences coding gene and transposon expression in embryos. By analyzing the RNA-seq

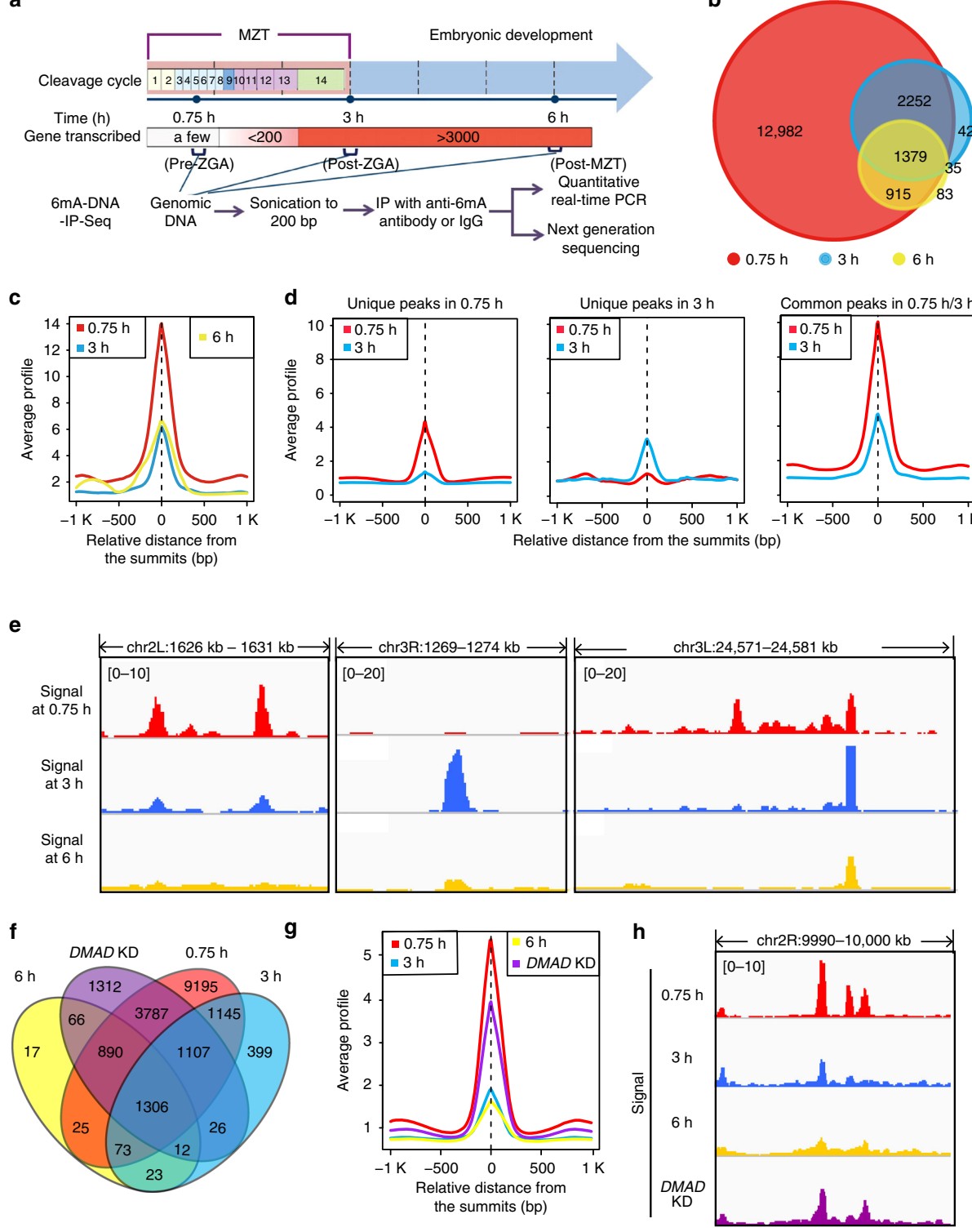

**Fig. 1** Dynamic distribution of 6mA in *Drosophila* early embryo genomes. **a** Timeline of the early embryogenesis and the 6mA-DNA-IP-Seq experimental process. The cleavage cycle is described according to a previous study[25]. DNA samples were collected from 0.75-h (nearly pre-MZT and pre-ZGA), 3-h (post-ZGA), and 6-h (post-MZT) post-fertilization embryos. **b** Overlap of 6mA enrichment peaks in 0.75-, 3-, and 6-h stage embryos. In cases where one peak in one sample overlapped multiple peaks in another sample, we selected the overlapped peak number from one sample as representative. **c** The average 6mA signal profiles for the common peaks in 0.75-, 3-, and 6-h stage embryos. **d** The average 6mA signal profiles in 0.75- and 3-h unique peaks and their common peaks. **e** Examples of 6mA-marked regions in which the 6mA modification signals were dynamically changed in early embryos. **f** Overlap of 6mA peaks identified in 0.75-, 3- and 6-h of wild type and 6-h *DMAD* knockdown (KD) samples. **g** The average 6mA signal profiles in peaks gained in 6-h *DMAD* knockdown samples, when compared with 6-h wild-type samples. These peaks were also detected in samples at 0.75-h or 3-h stages. **h** An example of 6mA signal in 0.75-, 3- and 6-h of wild-type stage and 6-h *DMAD* knockdown samples. MZT maternal-to-zygotic transition, ZGA zygotic genome activation

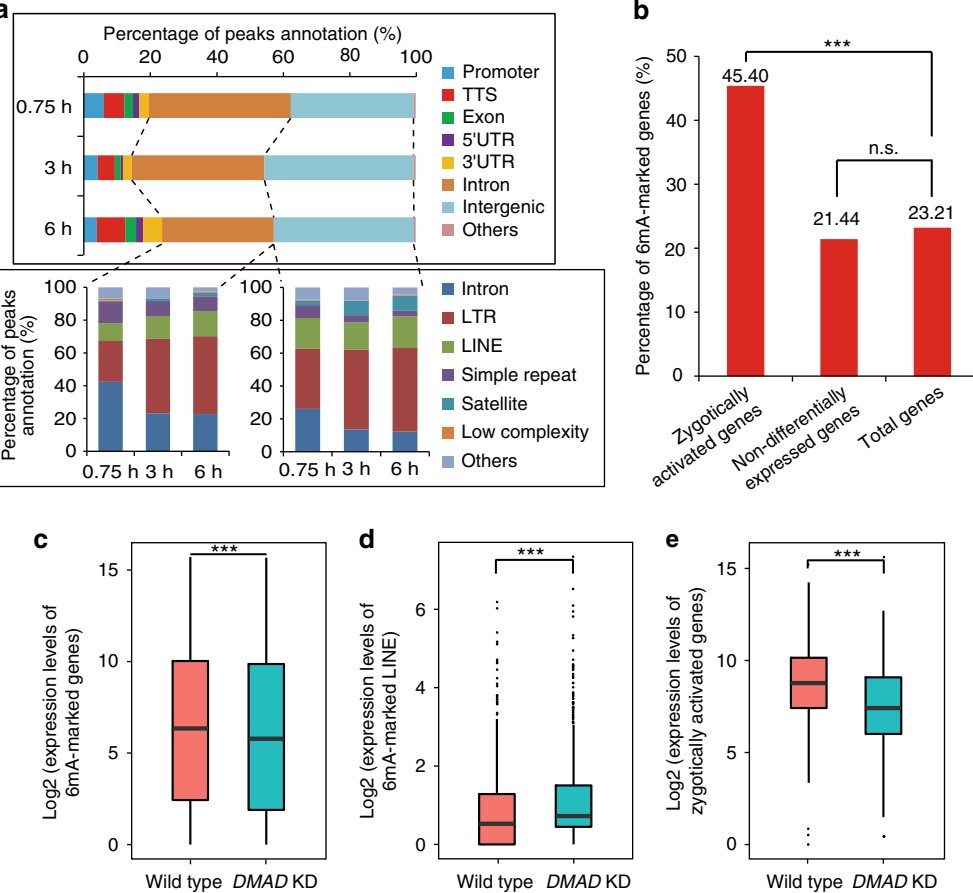

**Fig. 2** 6mA marks coding genes and associates with gene expression in early embryos. **a** Distribution of 6mA peaks on the *Drosophila* genome. Promoter, from 1 kb upstream to 100 bp downstream of the TSS of a gene; TTS, from 100 bp upstream to 1 kb downstream of transcription terminal site of a gene; Intergenic, 1 kb away from any gene. LTR and LINE are transposons. **b** Percentage of genes marked with 6mA. *P* value was calculated by one-tailed hypergeometric test. ***P < 0.001. **c–e** The expression of 6mA-marked genes (**c**), 6mA-marked LINEs (**d**), zygotically activated genes (**e**) in wild type and *DMAD* knockdown (KD) embryos. *P* values were calculated by one-tailed Student's *t* test. ***P < 0.001. For both boxplots, the centre line indicates the median, the bottom and top of the box show the first and third quartiles of the data, and the whiskers show the minimum and maximum values. Source data are provided as a Source Data file. TSS transcription start site, TTS transcription terminal site, n.s., not significant

data from *DMAD* knockdown and wild-type samples at the 6-h stage, we found that while the expression of LINE transposons was significantly up-regulated, the expression level of total 6mA-related coding genes was significantly downregulated in *DMAD* knockdown samples (Fig. 2c, d and Supplementary Figure 2e, f). Particularly, we noted that the expression of zygotically activated genes was significantly downregulated in *DMAD* knockdown sample (Fig. 2e and Supplementary Figure 2g). Thus, our findings suggest that 6mA modification associates with gene expression in early embryos.

**Identification of Jumu as a 6mA-DNA-binding protein**. To identify potential 6mA-DNA-binding proteins, we analyzed the DNA sequences of 6mA peak regions, and found that Forkhead domain (FHD)-binding motifs were highly enriched in the 6mA peak regions (Fig. 3a), raising a possibility that a Fox family protein binds to the 6mA-DNA and acts as a maternal factor to control early embryogenesis. We therefore performed affinity purification experiments using 6mA-modified DNA probes. Based on our 6mA data sets, we synthesized biotin-labeled DNA probes with or without 6mA modification and mixed probes with nuclear lysate from early embryos. Complexes immunoprecipitated by the biotin-labeled DNA probes were subjected to mass spectrometry analyses. Intriguingly, we found that Jumu, a Fox

family protein, was enriched in the immunoprecipitants purified by the 6mA-modified probes (Supplementary Figure 3a). We analyzed the expression level of all Fox family genes at 0.75-, 3- and 6-h, and found that *jumu* displayed the highest expression levels at all three stages (Supplementary Figure 3b). To test whether Jumu is a 6mA-DNA-binding protein, we performed western blot assays using an anti-Jumu antibody against purified complexes. As shown in Fig. 3b, complexes purified by the 6mA-modified probes exhibited a much higher Jumu signal than the control. We then performed gel-shift assays to measure the binding affinity of Jumu with 6mA-modified or unmodified DNA probes (FAM-labeled) that contained the FHD-binding sequence. The probe competition analysis revealed that Jumu preferentially bound the 6mA-modified DNA probe (Fig. 3c–e). To obtain further evidence, we measured the binding affinity of Jumu with 6mA-modified or unmodified probes by performing the surface plasmon resonance (SPR) kinetic assays, and found Jumu had much higher binding activity for 6mA-DNA (*K*d = 2.37 μM) than for unmodified DNA (*K*d = 42.1 μM) (Fig. 3f). Together, our findings suggest that Jumu preferentially binds the 6mA-modified DNA probes.

To rule out the preferential binding of Jumu with 6mA-modified DNA was attributed to the sequence specificity of the probes, we incubated the purified Flag-Jumu protein or the Flag-GFP protein (used as control) with genomic DNA from wild type

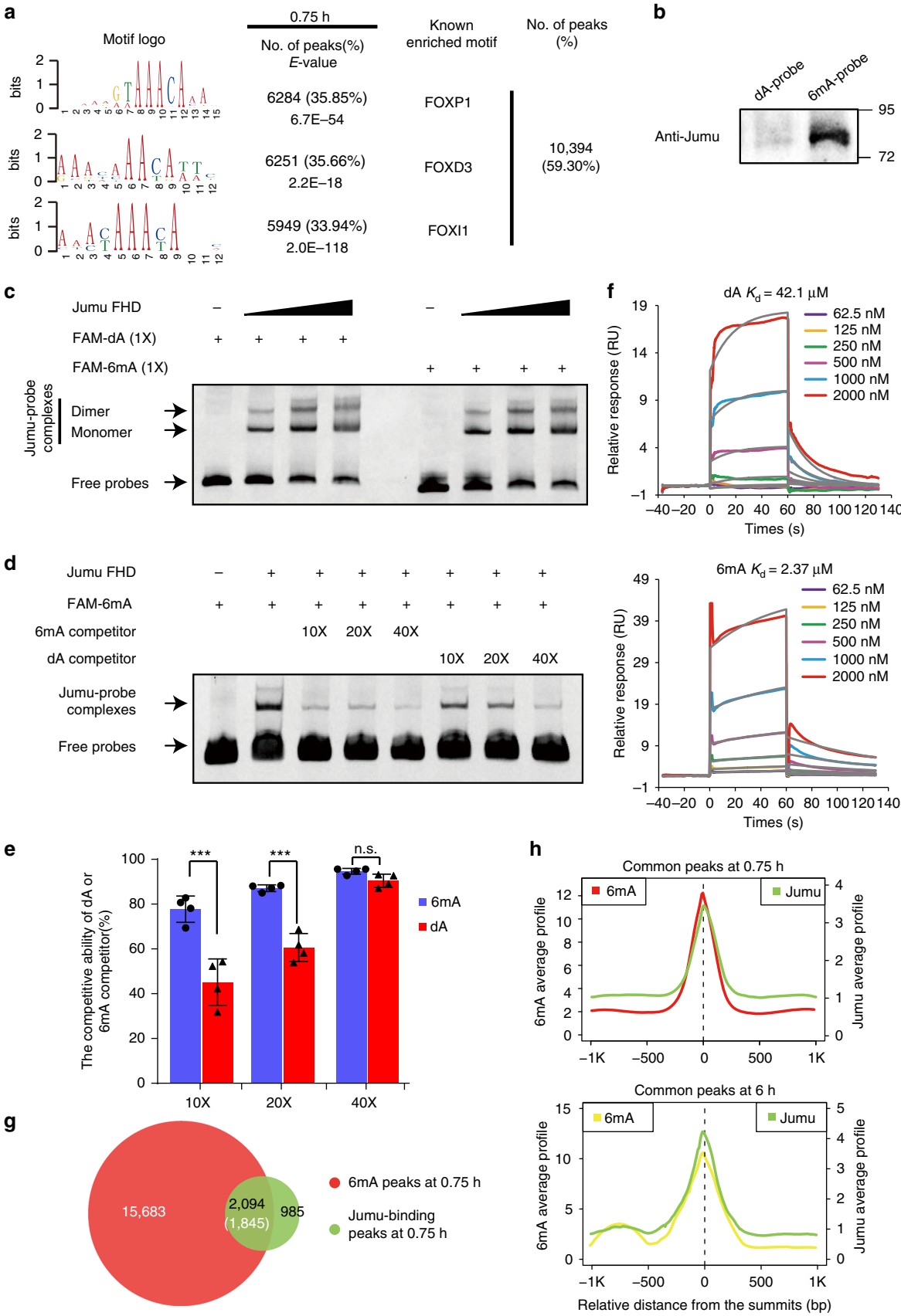

**Fig. 3** Jumu is a 6mA-marked DNA-binding protein. **a** Logo representations of Forkhead domain (FHD)-binding motifs identified in the 6mA peaks at the 0.75-h stage. For 6mA peaks in 0.75-h stage embryos, the number and percentage of peaks containing the motif and the E-value of motif occurrence were indicated. **b** Western blot assays using the anti-Jumu antibody against purified complexes. Complexes purified by the 6mA-containing DNA probes exhibited much higher Jumu signals than those purified by the control probes. **c** Electrophoretic Mobility Shift Assay (EMSA) showing the signals of the shifted probe bands in gradient dose of Jumu protein. The signals were progressively increased with gradient dose of Jumu protein present in the reactions. **d** Competitive EMSA in different titrations showing the different affinities of recombinant FHD of Jumu protein to dA-dsDNA and 6mA-dsDNA. **e** The competitive ratio of dA or 6mA competitor in different titrations by the quantitative intensity of bands boxed (**d**). The one-tailed Student's $t$ test was used to analyze statistical variance. Data expressed as means of 4 independent experiments. Error bars indicate mean ± s.d. ***$P < 0.001$. **f** Surface plasmon resonance (SPR) measurements of the interaction between recombinant FHD of Jumu protein and dA/6mA-dsDNA. Equilibrium and kinetic constants were calculated by a global fit to 1:1 Langmuir binding model. RU, resonance units. **g** Overlap of 6mA enrichment peaks and Jumu-binding peaks in 0.75-h embryos. **h** The average signal profile of 6mA and Jumu-binding in common 6mA/Jumu peaks in 0.75- and 6-h stage embryos. Source data are provided as a Source Data file. n.s. not significant

and *DMAD* mutant brains (with high level of 6mA). Complexes immunoprecipitated by anti-Flag beads were then subjected to dot blot assays. As shown in Supplementary Figure 3c, Jumu captured much more of the 6mA signal than the GFP control. We then performed high-throughput sequencing assays using genomic DNA from 0.75- and 6-h embryos captured by the Flag-Jumu protein. By performing MACS2 analysis, we identified 3,079 and 921 Jumu-binding sites in the 0.75- and 6-h embryos, respectively. Strikingly, 68 and 78% of Jumu-binding peaks overlapped with the 6mA-enriched peak regions in the 0.75- and 6-h embryos, respectively (Fig. 3g and Supplementary Figure 3d), and the summits of the Jumu-binding peaks almost overlapped with those of the 6mA peaks (Fig. 3h). Importantly, the signal strength of 6mA in the 6mA/Jumu common peaks was much stronger than in the unique 6mA peaks (Fig. 3h and Supplementary Figure 3e). These results suggest that Jumu is a 6mA-marked DNA-binding protein.

**Jumu functions as a maternal factor in early embryos.** Based on the above findings, we speculated that Jumu acts as a maternal factor to regulate early embryogenesis by binding 6mA mark. We thus examined abundance of Jumu protein at various embryonic stages. As shown by western blot assays, the Jumu was present at relatively high levels in early embryos, but its levels were reduced at the 10-h stage (Fig. 4a). Of note, Jumu was also found at considerable levels in unfertilized eggs and ovaries (Fig. 4a), suggesting that Jumu were preloaded into mature oocyte. We then employed the CRISPR/Cas9 system to generate two *jumu* mutant alleles, *jumu*[1] and *jumu*[2] (Fig. 4b). Genetic tests showed that about 45% ($n = 116$) of zygotic *jumu* mutants finally developed into adult flies, in which Jumu expression was completely abolished (Fig. 4c). Of note, although loss of zygotic Jumu reduced female fertility by affecting germline development, zygotic *jumu* mutant females still produced a few eggs, in which the maternal Jumu was absent (Fig. 4d). To determine the maternal role of Jumu, we used maternal *jumu*[(M−)] mutant eggs to perform further genetic experiments. As shown in hatching rate analysis, compared with the wild-type control, ~72% of *jumu*[(M−,Z+)] embryos failed to hatch into larvae, when maternal *jumu* mutant eggs were fertilized by the wild-type sperm (Fig. 4e and Supplementary Figure 4a). Cuticle analysis showed that about 30% of dead mutant embryos (143/482) had displayed segmentation pattern with different numbers of segments, whereas about 70% of dead mutant embryos (339/482) even showed no segmentation phenotype (Fig. 4f). Further Immunostaining assays revealed that compared to the wild-type control, the hexagonal-actin network was apparently disorganized in about 50% (132/265) maternal *jumu* mutant embryos at the nuclear cycle 14 (Fig. 4g). Additionally, nuclear fallout phenotypes were observed in some *jumu*[(M−,Z+)] mutant embryos (Fig. 4g and Supplementary Figure 4b). These findings suggested a role of maternal Jumu

in regulating cellularization in the early embryos. We noted that ~86% of *jumu*[(M−,Z−)] mutant displayed embryonic lethal phenotypes (Fig. 4e and Supplementary Figure 4a). Only a few of the *jumu*[(M−,Z−)] mutant (~1.7%, 33/1934) reached adulthood. Notably, the *jumu* mutant sperm was sufficient to support normal embryogenesis when it was used to fertilize wild-type eggs (Fig. 4e and Supplementary Figure 4a). Because *jumu*[(M−,Z−)] embryos displayed severer embryonic lethal phenotypes than that in *jumu*[(M−,Z+)] embryos, we examined the expression of Jumu in *jumu*[(M−,Z+)] embryos at 0–1 and 2–4 h stages, and found Jumu was zygotically expressed in embryos (Supplementary Figure 4c). These results suggest that Jumu plays important maternal roles during embryogenesis.

**Maternal Jumu regulates early embryonic gene expression.** Next, we tested whether the maternal Jumu controls the proper gene expression in early embryos, and generated RNA-seq data sets from *jumu*[(M−,Z+)], *jumu*[(M−,Z−)] and wild-type embryos at the 3-h stage. Because loss of maternal Jumu leads to embryonic lethal phenotypes, we first compared the gene expression profile of *jumu*[(M−,Z+)] or *jumu*[(M−,Z−)] mutant embryos with that of wild-type embryos, and identified 2198 genes that were significantly up/downregulated in both *jumu*[(M−,Z+)] and *jumu*[(M−,Z−)] mutant embryos (Fig. 5a). We defined these 2,198 genes as genes significantly regulated by the maternal Jumu (hereafter referred to as GSRJ), and divided the GSRJ into two groups, group 1 (up-regulated genes) and group 2 (down-regulated genes). Gene ontology (GO) analysis revealed that some of GSRJ genes were development-related (Supplementary Figure 5a). Thus, lethality of maternal *jumu* mutant embryos could be, at least in part, attributed to abnormal expression of development-related genes. Of note, a number of the GSRJ genes encoding maternal transcription factors (e.g., Zelda[20] and Lilli[26]) were found to be up-regulated in 3-h *jumu*[(M−,Z−)] embryos. These findings suggest that loss of maternal Jumu changes gene expression pattern in early embryos.

**Jumu regulates 6mA-marked *zelda* in early embryos.** Given that Jumu binds to 6mA-marked DNA, we next investigated the role of Jumu in regulating its binding-target genes. We compared the Jumu-bound genes (definition seen in Methods) with the GSRJ, and found 189 Jumu-bound GSRJ (Fig. 5b). The remaining GSRJ without Jumu-binding sites were referred to as the Jumu-indirect GSRJ. By scanning the 6mA peaks around the Jumu-bound GSRJ, we found that 95% (180/189) of Jumu-bound GSRJ were marked by 6mA, whereas only about 27% (534/2,009) of Jumu-indirect GSRJ were marked by 6mA (Fig. 5c).

To understand molecular mechanisms by which Jumu regulates early embryogenesis, we carefully examined the 189 Jumu-bound GSRJ, and found that some important genes, such as *zelda*, *lilli*, *lola* and *Trf2*, were marked by both 6mA and

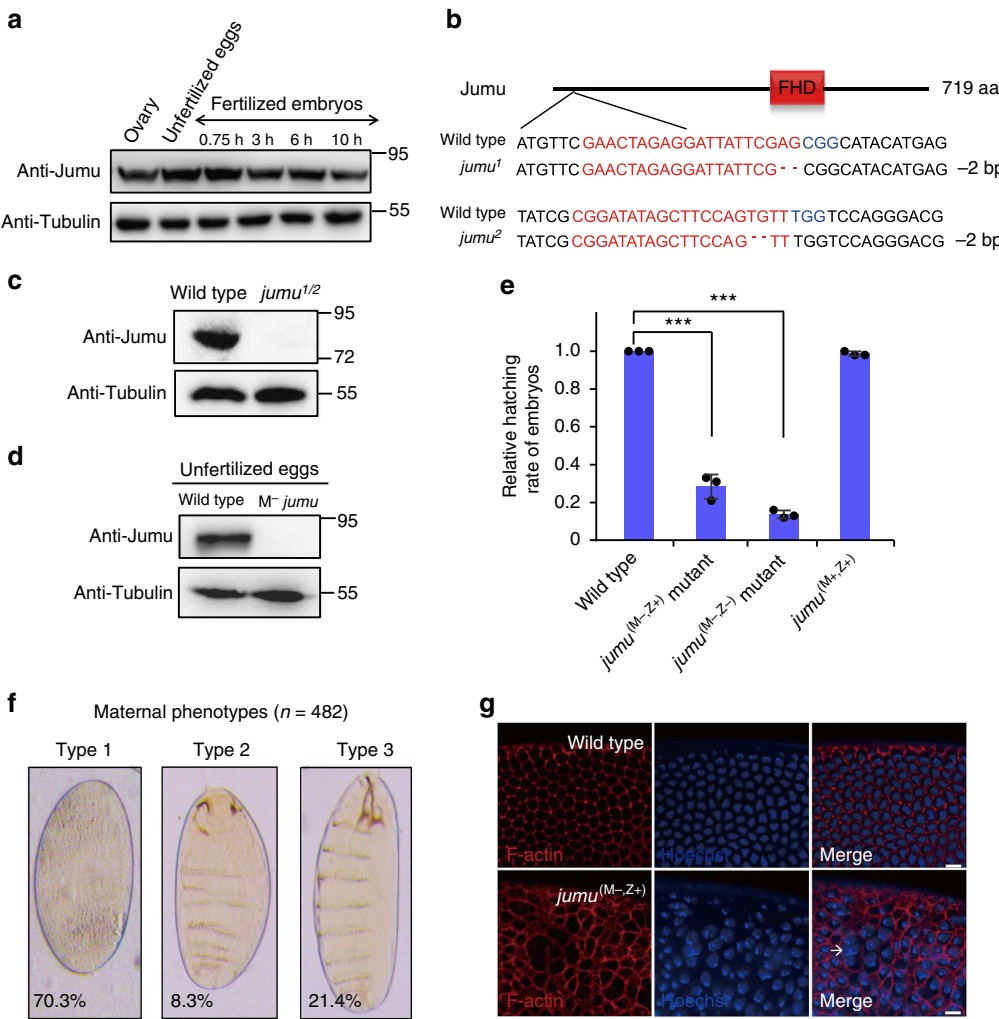

**Fig. 4** Jumu functions as a maternal factor to regulate embryogenesis. **a** Western blot assays showing Jumu protein abundance in ovary, unfertilized eggs and embryos at various embryonic stages as indicated. **b** Two *jumu* mutant alleles, *jumu¹* and *jumu²*, were generated by the CRISPR/Cas9 system. **c, d** Western blot assays were performed to show expression levels of Jumu protein in wild type or *jumu* mutant ovaries (**c**), and in wild type or maternal *jumu* mutant eggs (**d**). **e** Relative hatching rate of embryos with indicated genotypes. The two-tailed Student's *t* test was used to analyze statistical variance. Data expressed as means of 3 independent experiments. Error bars indicate mean ± s.d. ***$P < 0.001$. **f** Three typical phenotypes were observed in *jumu* maternal lethal embryos (M⁻, Z⁺, *jumu*). **g** Wild type and maternal *jumu* mutant embryos were stained with phalloidin and Hoechst to visualize F-actin (red) and DNA (blue), respectively. Scale bar, 10 μm. Source data are provided as a Source Data file

Jumu-binding signals (Fig. 5d and Supplementary Figure 5b). Among these, we considered the *zelda* gene as being of particular interest because its product Zelda acts as a pioneer transcription factor essential for MZT[20]. In our RNA-seq data sets, *zelda* was expressed at considerable levels in 3-h wild-type embryos, but was further up-regulated in maternal Jumu mutant (Supplementary Figure 5c). To validate the role of 6mA, we generated maternal DMAD mutant embryos (see Methods), and found that loss of the maternal DMAD caused embryonic lethal (Supplementary Figure 5d, e). Moreover, *zelda* was down-regulated in maternal DMAD mutant (Supplementary Figure 5f, g). Thus, our findings suggest that Jumu negatively regulates *zelda* expression in early embryos likely through 6mA modification.

We then examined embryos with overexpression of maternal Zelda. Consistent with previous findings[22], Zelda overexpression caused partial embryonic lethal (710/1374), and some Zelda overexpression embryos displayed celluarization defects (Supplementary Figure 6a–e). Because *zelda* was up-regulated in maternal Jumu mutant embryos, we tested whether the embryonic lethal phenotype in the maternal Jumu mutant is attributed to abnormal expression of Zelda target genes. We generated the RNA-seq dataset of embryos (3-h stage) with overexpression of maternal Zelda. By integrating this dataset with the Zelda ChIP-seq data[27], we identified 3,165 Zelda target genes. Notably, 78% (1715/2,198) of the GSRJ were Zelda-target genes, and 77% (146/189) of Jumu-bound GSRJ were Zelda-target genes (Fig. 5e). Further analyses revealed that the group 1 genes of GSRJ up-regulated in maternal Jumu mutant were up-regulated in Zelda overexpression samples, and likewise, the group 2 genes of GSRJ downregulated in maternal Jumu mutant were downregulated in Zelda overexpression samples (Fig. 5f, g). To obtain biological evidence, we generated *zelda* knock-down females carrying different transgene combinations, P{*nos-gvp*}²ᵉᵈ/P{*uas-zelda-RNAi*} and P{*uasp-artmiR-zelda*}/P{*nos-gvp*}³ᵉᵈ, which produced embryos with partial and strong knockdown of maternal Zelda, respectively (Fig. 6a and Supplementary Figure 6f). Consistent with previous findings[20], strong knockdown of maternal Zelda caused complete embryonic lethal (Supplementary Figure 6g), whereas partial

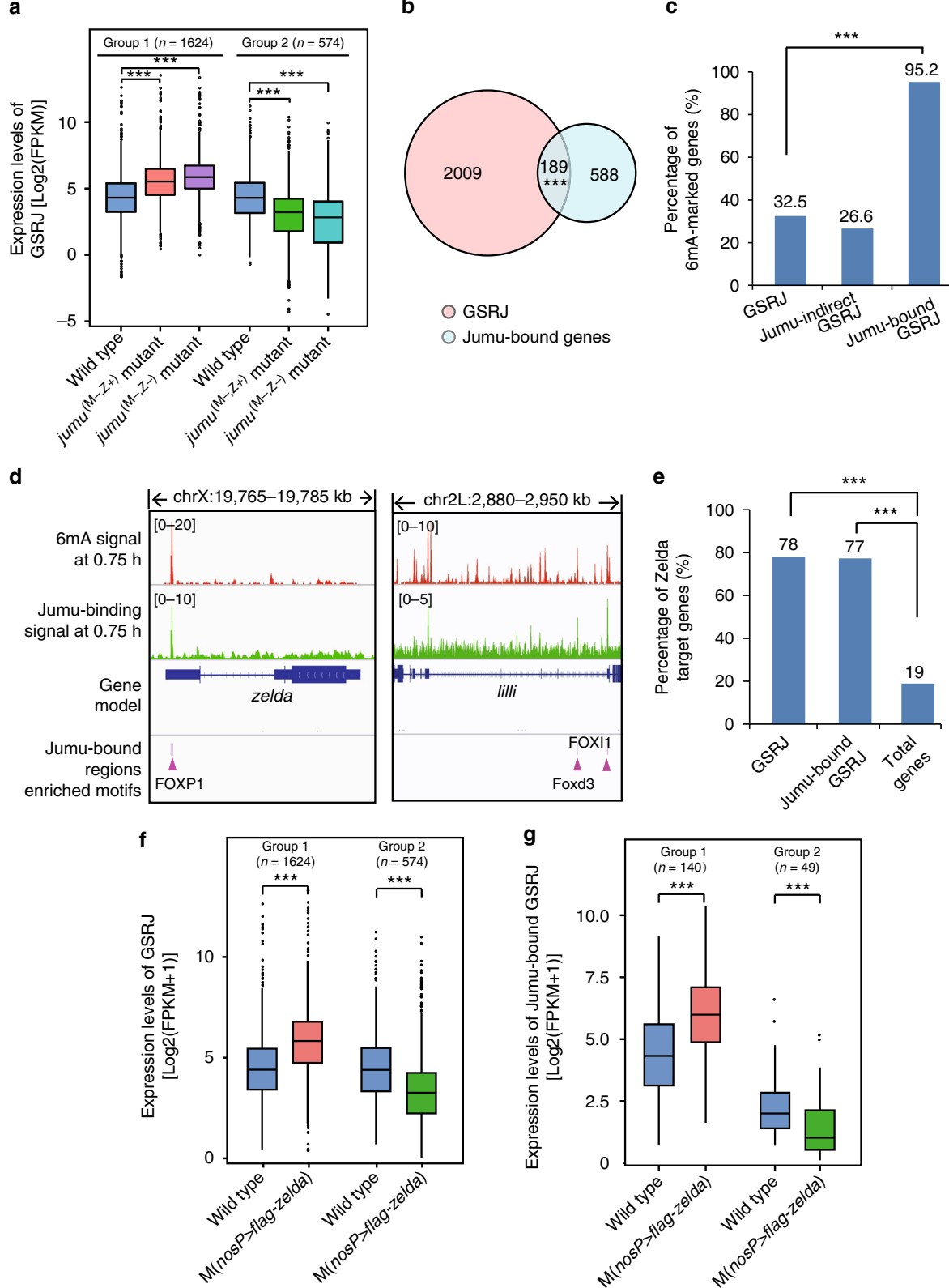

knockdown of maternal Zelda did not significantly affect embryonic development (Fig. 6b, c and Table 1). Notably, partial knockdown of maternal Zelda suppressed the embryonic lethal phenotype induced by loss of maternal *jumu* (Fig. 6b, c and Table 1). Thus, our findings suggest that Jumu regulates early embryogenesis, at least in part, through controlling 6mA-marked Zelda.

## Discussion

The precise activation of zygotic genomes is essential for the normal embryogenesis; however, the molecular mechanisms that regulate this process have remained poorly understood. Our recent study showed that 6mA dynamics in *Drosophila* early embryos almost corresponds to the process of MZT, raising a possibility that 6mA forms an epigenetic mark contributing to

**Fig. 5** Jumu regulates embryonic gene expression via 6mA modification. **a** A total of 1624 genes of GSRJ (group 1) were significantly up-regulated and 574 genes of GSRJ (group 2) were significantly downregulated in *jumu* mutant embryos. *P* values were calculated by one-tailed Student's *t* test. ***$P < 0.001$. GSRJ, genes significantly regulated by the maternal Jumu. **b** Overlap of Jumu-bound genes and GSRJ. *P* value ($P = 7.3e\text{-}06$) was calculated by one-tailed hypergeometric test. ***$P < 0.001$. **c** Percentage of 6mA-marked genes in different gene sets as indicated. *P* value was calculated by one-tailed hypergeometric test. ***$P < 0.001$. **d** Examples for 6mA and Jumu-binding signals around important gene (*zelda, lilli*). The enriched motifs in Jumu-bound regions were shown in figures. **e** Percentage of Zelda target genes in different gene sets as indicated. *P* value was calculated by one-tailed hypergeometric test. ***$P < 0.001$. **f** A total of 1624 genes of GSRJ (group 1) were upregulated and 574 genes of GSRJ (group 2) were downregulated in *zelda* overexpressed embryos. *P* values were calculated by one-tailed Student's *t* test. ***$P < 0.001$. **g** Jumu-bound group 1 GSRJ were upregulated and Jumu-bound group 2 GSRJ were downregulated in *zelda* overexpressed embryos. *P* values were calculated by one-tailed Student's *t* test. ***$P < 0.001$. For all boxplots, the centre line indicates the median, the bottom and top of the box show the first and third quartiles of the data, and the whiskers show the minimum and maximum values. Source data are provided as a Source Data file

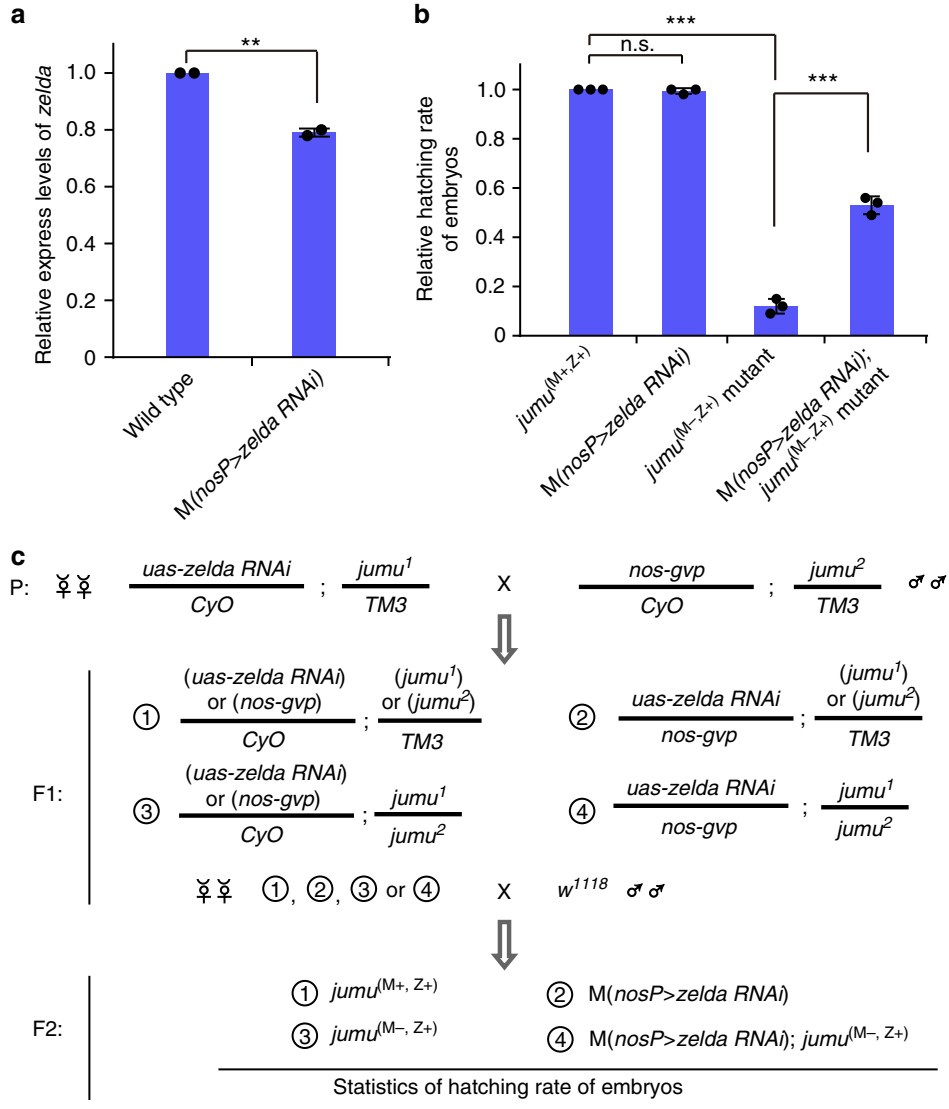

**Fig. 6** Maternal Zelda overexpression causes partial embryonic lethal. **a** Quantitative RT-PCR experiments were used to analyze the effectiveness of *zelda* knockdown in 0.75-h stage embryos. The two-tailed Student's *t* test was used to analyze statistical variance. Data expressed as means of 2 independent experiments. Error bars indicate mean ± s.d. **$P < 0.01$. **b** Relative hatching rate of embryos with indicated genotypes. The two-tailed Student's *t* test was used to analyze statistical variance. Data expressed as means of 3 independent experiments. Error bars indicate mean ± s.d. ***$P < 0.001$. **c** Females with indicated genotype (*uas-zelda RNAi/CyO; jumu$^1$/TM3*) were crossed with males (*nos-gvp/CyO;jumu$^2$/TM3*), the eggs produced by female progenies were then fertilized by the wild-type sperm. Unhatched embryos and larvae with indicated genotypes were counted separately. Source data are provided as a Source Data file. n.s. not significant

### Table 1 Hatching rate of embryos with indicated genotypes

| Genotype of embryos | Hatching rate | | |
| --- | --- | --- | --- |
| | Replicate 1 | Replicate 2 | Replicate 3 |
| jumu(M+, Z+) | 0.91 (480/530*) | 0.90 (285/317*) | 0.86 (425/494*) |
| M (nosP > zelda RNAi) | 0.91 (1018/1114*) | 0.90 (130/145*) | 0.84 (323/383*) |
| jumu(M−, Z+) mutant | 0.11 (32/296*) | 0.14 (11/81*) | 0.08 (27/343*) |
| M (nosP > zelda RNAi); jumu(M−, Z+) mutant | 0.51 (338/668*) | 0.49 (512/1047*) | 0.42 (286/681*) |

*Sample size

MZT. In this study, we shed light on this intriguing issue, and found 6mA can mark and associates with zygotic gene expression in early embryos. Importantly, we found that the Fox family protein Jumu bound 6mA-marked gene and functioned as a maternal factor to control the proper activation of zygotic genomes. Thus, our findings support a notion that 6mA-formed epigenetic code can be read by the maternal factor, thus controlling normal embryogenesis.

It has been suggested that Zelda and Zelda-like proteins function as pioneer transcription factors to initiate zygotic genome activation[20]. Interestingly, we found that zelda was marked with 6mA and regulated by Jumu. Bioinformatics analysis showed that 78% (1715/2,198) of the GSRJ were also Zelda's targets. Consistently, genetic experiments showed that maternal overexpression of Zelda caused embryonic lethal phenotype, which mimics that observed in Jumu maternal mutant embryos. Importantly, we found that partial knockdown of Zelda significantly suppressed the embryonic lethal phenotype induced by loss of maternal Jumu. These findings strongly suggested that Zelda is one of critical target genes of Jumu, and that Jumu regulates the proper zygotic genome activation, at least in part, through regulating Zelda (Fig. 7a). Previous studies have suggested that loss of maternal Zelda leads to either down-regulation or upregulation of target genes in Drosophila embryos, suggesting that Zelda regulates gene expression in either direct or indirect manner, although Zelda is a positive regulator of gene expression. For the case of indirect targets, Zelda could activate a set of miRNAs, which inhibit sets of downstream target genes[20,28]. In addition to zelda, our results suggest that other important genes, such as lilli and lola, were marked with 6mA and regulated by Jumu. Based on our data analysis, we propose a model by which the maternal factor Jumu specifically reads a 6mA-based epigenetic code to control other transcription factors including Zelda, thereby ensuring the proper embryogenesis (Fig. 7b). Of note, Jumu is required for both germline development and embryogenesis; however, the actions of Jumu appear to be different between two biological contexts (Supplementary Figure 7a–g). In the context of early embryos, loss of maternal Jumu led to upregulation of Zelda, and loss of Jumu and overexpression of maternal Zelda caused similar embryonic lethal phenotypes. Importantly, transcriptomic profiles of GSRJ in maternal Jumu mutant were very similar to that in maternal Zelda overexpression samples. By contrast, in context of germline, knockdown or overexpression of Zelda in germ cells had no apparent effects on the normal oogenesis, suggesting that Zelda is dispensable for germline development (Supplementary Figure 7h).

Here we identified a Fox family protein Jumu functions as a 6mA reader to regulate gene expression. However, it still remains unknown about whether 6mA modification has a role to inhibit the binding of protein–DNA. Nevertheless, because Fox family proteins have conserved roles in controlling embryonic development and tissue homeostasis, misregulation of some Fox family proteins has been implicated in many human diseases including cancers[29,30]. It will be interesting to investigate how Fox family proteins function in concert with 6mA epigenetic mark to regulate development in mammals and in human diseases.

## Methods

**Drosophila strains.** Fly stocks were maintained under standard culture conditions. The Drosophila w1118 strain was used for collecting the wild-type embryos in this study. DMAD null flies, DMAD1 and DMAD2, were described previously[11]. The following strains were generated in this study: (1) Two jumu mutant alleles, jumu1 and jumu2, were generated by the following method[11]. Briefly, the target DNA sequence selected for CRISPR RNA-guided Cas9 nuclease was predicted by crisper/cas target software (http://zifit.partners.org/ZiFiT). The templates for gRNA in vitro transcription were amplified from pMD19T- gRNA scaffold. Then the gRNA was in vitro transcribed through run-off reactions and injected into vasa-Cas9 embryos. The flies were incubated under standard culture conditions and outcrossed for further screening by T7E1 enzyme digestion and sequencing; (2) The transgene line, P{uasp-flag-zelda}, in which the full-length zelda was inserted into UASp-flag vectors; (3) The zelda knockdown line, P{uasp-artmiR-zelda} was generated by the following method. A pair of designed primers was denatured and annealed to form short dsDNA, which were then digested by NheI and EcoRI enzymes to insert into UASp BX vectors. The w1118 strain was used as the host for P element-mediated transformations. P{uas-zelda RNAi} was obtained from Tsinghua fly center. P{nos-gvp}2ed and P{nos-gvp}3ed were used as a maternal driver. The detailed information of primers was described in the Supplementary Table 1.

**Embryo preparation.** The flies were maintained in large population cages in an incubator at standard conditions (25 °C). As the protocol for collecting embryos, the well-fed flies were used to lay eggs in bottles, each of which was covered by a petri dish with agar gel. In this work, six students and postdocs carefully collected and examined embryos under the light microscopes. For the 0.75-h stage, embryos were collected for 30 min, and then allowed to develop for 15 min. Older embryos were removed using the light microscopes[27] and the embryos were cleaned using washing buffer (1x PBS) to avoid contamination. All actions were stopped at the 0.75-h, at which the samples were immediately put into liquid nitrogen and stored at −80 °C. Likewise, the ranges of 3-h and 6-h are 2.5–3 h and 5.5–6 h, respectively.

**Purification of genomic DNA.** Drosophila samples were collected and dissected with indicated stages and tissues. Genomic DNA was extracted using a Wizard genomic DNA purification kit (Promega) following the manufacturer's instructions.

**6mA-DNA-IP-Seq.** Purified genomic DNA was sonicated to 200 base pairs (bp). 5 μg of fragmented genomic DNA was immunoprecipitated with 5 μg of rabbit 6mA antibody (Abcam) or rabbit IgG overnight at 4 °C in a final volume of 500 μl immunoprecipitation buffer (10 mM sodium phosphate, pH 7.0, 140 mM NaCl, 0.05% Triton X-100). The mixture was incubated with 50 μl protein A/G agarose beads for 2 h at 4 °C and then washed three times with 1 ml of immunoprecipitation buffer. The beads were then treated with proteinase K and the methylated DNA was purified by phenol–chloroform extraction followed by ethanol precipitation for library construction. End repair, 3′-adenylation, adaptor ligation, and PCR amplification were performed according to the Illumina TruSeq DNA sample preparation procedures. The libraries were sequenced using an Illumina HiSeq 2500 V4 platform. Of note, the anti-6mA used in this study can efficiently capture either single-strand or double-strand genomic DNA (Supplementary Figure 1b, c).

It is worth to note that both IgG- and input-based controls have been widely used to call peaks in DNA-IP-seq experiments in many studies[31,32]. In this study, we found that there was no significant bias between two approaches. For example, at the 0.75-h stage, we found that 96% of Input-controlled 6mA peaks were overlapped with IgG-controlled 6mA peaks (Supplementary Figure 1f).

**RNA sequencing (RNA-seq).** Total RNAs were isolated from embryos at the indicated stages using Trizol reagent (Thermo Fisher). The mRNAs were purified using a Dynabeads mRNA DIRECT Kit (Thermo Fisher) according to the user

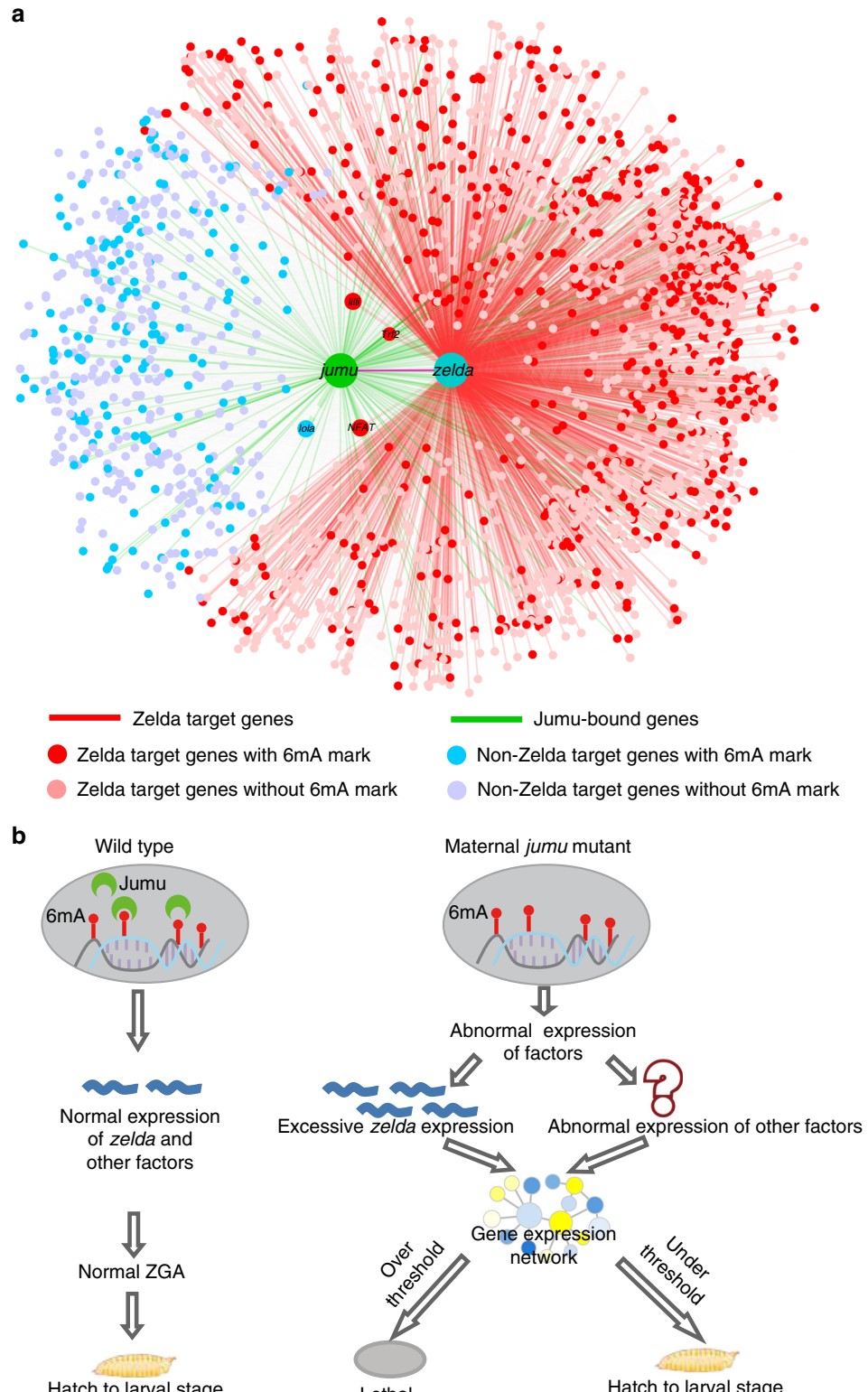

**Fig. 7** Regulatory gene network of GSRJ mediated by Jumu. **a** The regulatory network by which Jumu regulates embryonic gene expression in Zelda-dependent and Zelda-independent manners. The Jumu/Zelda target genes were as defined in the main text. **b** Model for Jumu as a 6mA reader that regulates embryonic gene expression. ZGA, zygotic genome activation. Source data are provided as a Source Data file

manual. Illumina sequencing libraries were constructed according to the manufacturer's instructions and sequenced using an Illumina HiSeq 2500 V4 or NovaSeq platform.

**Protein-DNA interaction and high-throughput sequencing.** Plasmids were transfected into HEK293T cells using polyethyleneimine (PEI, Sigma). The cells

were treated with lysis buffer (50 mM Tris-HCl, pH 7.4, 500 mM NaCl, 1% NP-40, protease inhibitor) after culturing for 48 h. Following centrifugation (20,000 × g) for 10 min at 4 °C, the supernatants were mixed with anti-FLAG M2 affinity beads (Sigma) and incubated for 3-h at 4 °C. Then the beads were washed twice with lysis buffer and twice with TBS (20 mM Tris-HCl, pH 7.4, 150 mM NaCl). Five micrograms of fragmented genomic DNA and immunoprecipitation buffer to a

final volume of 500 μl was added to the beads and held overnight at 4 °C. Then the mixture was washed three times with 1 ml immunoprecipitation buffer and the beads were treated with proteinase K. The bound DNA was purified by phenol–chloroform extraction followed by ethanol precipitation. Sequencing libraries were prepared according to the Illumina TruSeq DNA sample preparation procedures and high-throughput sequencing was performed using an Illumina HiSeq 2500 V4 platform.

**Synthesis of DNA probes**. Unlabeled or biotin/FAM-labeled dA/6mA dsDNA probes were synthesized using PCR reactions with 2′-deoxyadenosine-5′-triphosphate (dATP) or N6-methyl-2′-deoxyadenosine-5′-triphosphate (dm$^6$ATP) (Trilink) in place of dATP. The PCR products were purified using Universal DNA Purification Kit (Tiangen). The sequences of the dsDNA substrates are listed below:

Probe. 5′-GCTATTTATGCGTTCTTTTCTTCTTTCTCATTTCTATATTTATAGTGTGTGTGTGTGTGGGTGTGTTTGTTGTATGTGGTTAAATATGCTTCGTTTTTGTTTAATTGGTTTTTGTTTTTGTGATGATCATGGG-3′

**DNA pull-downs and Mass Spectrometry**. Embryos were homogenized in STMDPS-250mM buffer (0.25 M sucrose, 25 mM KCl, 5 mM MgCl$_2$, 50 mM Tris-HCl, pH 7.4, 1 mM DTT). After 75-μm filtration, the crude nuclear pellet was obtained by centrifugation at $800 \times g$ for 10 min. The pellet was then resuspended in STMDPS-1.8 M buffer (1.8 M sucrose) layered over STMDPS-2M buffer (2 M sucrose) and centrifuged at $100,000 \times g$ for 1 h. The nuclei were lysed in nuclear lysis buffer (420 mM NaCl, 20 mM HEPES, pH 7.9, 20% v/v glycerol, 2 mM MgCl$_2$, 0.2 mM EDTA, 0.1% NP40, complete protease inhibitor without EDTA (Roche) and 0.5 mM DTT) and centrifuged at $100,000 \times g$ for 1 h. For each DNA pull-down, 10 μg of biotin-labeled dA/6mA dsDNA probes was immobilized on 60 μl of streptavidin beads (GE) by incubating for 1 h at room temperature in DNA-binding buffer (10 mM sodium phosphate, pH 7.0, 140 mM NaCl, 0.05% NP40). The beads were then incubated with 400 μg of nuclear extract in a total volume of 600 μl of protein binding buffer (50 mM Tris-HCl, pH 8, 150 mM NaCl, 1 mM DTT, 0.25% NP40 and complete protease inhibitors) for 2 h. After washing three times with protein binding buffer, proteins were eluted and digested using Trypsin (Promega), and then subjected to LC-MS/MS assays. The resulting MS/MS data were processed using Thermo Proteome discovery (version 1.4.1.14), and the tandem mass spectra were searched against the UniProt *Drosophila* proteome database[33].

**Electrophoretic mobility shift assay (EMSA)**. EMSAs were performed using both 5′ and 3′-FAM-labeled dsDNA probes and GST-Jumu FKH domain protein (amino acid residue 391–540). Briefly, the GST-Jumu FKH protein was expressed in *Escherichia coli* strain BL21 (DE3) and purified by GST affinity purification. The FAM-dA/6mA dsDNA probes (15 nM) were incubated with a gradient dose of 1, 2 and 4 μM GST-Jumu FKH protein in buffer containing 20 mM HEPES, pH 7.5, 50 mM KCl, 1 mM DDT, 1 mM EDTA, 0.01 mg/ml BSA, 2% ficoll 400 and 5 ng/μl Poly (dI:dC) at 25 °C for 20 min. For competitive EMSAs, 2 μM GST-Jumu FKH protein and indicated molar excess of the competitive dA/6mA dsDNA (10, 20 and 40-fold) were used. Reaction products were subjected to 6% native polyacrylamide gel and run for1h at 80 V in 0.5 × TBE buffer. The gels were scanned by Typhoon FLA 9500 (GE Healthcare). The quantitation of band intensity in 3e was measured using ImageJ software.

**Surface plasmon resonance (SPR) measurements**. Biotinylated-dA/6mA DNA probes were captured on SA sensor chip (91–97 response units). Another blank flow cell was used as reference to correct for instrumental and concentration effect. SPR experiments were performed using Biacore 8 K instrument (GE Healthcare) in running buffer containing 20 mM Tris-Cl, pH 8.0, 150 mM NaCl and 0.005% Tween 20 at 25 °C. Jumu FKH proteins with increasing concentrations were injected into the dA/6mA probes surface and blank flow cell for 1 min at a flow rate of 30 μl/min, dissociated for 1 min in running buffer. Equilibrium and kinetic constants were calculated by a global fit to 1:1 Langmuir binding model (Biacore 8 K evaluation software).

**Western blot analysis**. Western blots were performed by using standard protocols. The following reagents were used: mouse anti-β-Tubulin (1:2,000; Cwbio); anti-Flag M2 antibody (1:5,000; Sigma); mouse anti-Jumu (1:1000). The antibody against Jumu was generated by immunizing mice with a recombinant protein GST-Jumu (amino acids 460–700) that was produced in *E. coli*. Uncropped blots are provided in Source Data file.

**Dot blot assay**. Different DNA samples were denatured at 95 °C for 10 min and spotted on nitrocellulose membranes. DNA was cross-linked to the membranes by UV irradiation and the membranes were blocked in 5% BSA in PBS containing 0.5% Tween 20 (PBST) for 1 h at room temperature. The membranes were then incubated with a 1:10,000 dilution of a 6mA antibody (Abcam) overnight at 4 °C. After three rounds of washes with PBST, the membranes were incubated with a 1:5,000 dilution of HRP-conjugated anti-rabbit IgG secondary antibody. The

membranes were then washed with PBST three times and treated with enhanced chemiluminescence. Uncropped blots are provided in Source Data file.

**Analysis of hatching rate and cuticle preparation of embryos**. Embryos were collected for 24 h at 25 °C, then removed from the adults and allowed to develop for another 36 h. Then, unhatched embryos and larvae were counted separately. For cuticle preparation, unhatched embryos were washed and dechorionated with 50% bleach. Dechorionated embryos were mounted on slides with Hoyer's medium, and baked at 60 °C overnight. Embryos were viewed and scored using a Leica MZ16 stereo zoom microscope.

**Quantitative Real-Time PCR Analysis**. Embryos were collected at the indicated stages for RNA extraction. Total RNA was extracted using Trizol reagent (Thermo Fisher) and cDNA was generated with a FastQuant RT Kit (with gDNase) (Tiangen). Real-time quantitative PCR reactions were performed in triplicate on a LightCycler_480 Real-time PCR instrument (Roche) using the UltraSYBR mixture (Cwbio). *Actin*5C was used as the constitutive control for normalization of candidate gene expression. Relative gene expression was calculated using the ΔΔCt method following the manufacturer's instructions. The qRT-PCR primers for RNA quantification are listed in the Supplementary Table 1.

**Embryo staining**. Embryos were fixed in fixation buffer (4% formaldehyde and 0.3% Tween-20 in PBS) for 30 min and washed in PBST (0.3% Tween-20 in PBS) for 15 min. F-actin and DNA were visualized by Alexa Fluor™ 546 Phalloidin (1:30, Thermo Fisher) and by Hoechst (1:2,000, Sigma), respectively. They were added to PBST and incubated with embryos overnight at 4 °C, followed by washing for three times (10 min per time) in PBST. The images were collected on a Zeiss LSM 710 Meta confocal microscopy.

**Gene knockdown in *Drosophila* embryos**. dsRNA fragments corresponding to *DMAD* and *gfp* mRNAs were synthesized in a PCR reaction and then fused to the T7 RNA polymerase binding site at both 5′ and 3′ ends. These were then used to generate dsRNA in vitro using the RiboMAX™ Large Scale RNA kit (Promega) following the manufacturer's instructions. The dsRNA (1 μg/μl) was injected into $w^{1118}$ embryos. The embryos were incubated at room temperature for turnover of the target protein. The primers used for the generation of *DMAD* and *gfp* DNA fragments are shown in the Supplementary Table 1.

**6mA-DNA-IP-Seq data analysis for *Drosophila***. After performing DNA immunoprecipitation (DNA-IP) experiments using an anti-6mA antibody, 125 bp paired-end high-throughput sequencing was performed on an Illumina HiSeq 2500 V4 platform by Berry Genomics company. In total, 15.6, 11.7, 11.1, 6.8, 9.2 and 15.7 million paired-end reads were obtained from 0.75-h 6mA IP and IgG samples, 3-h 6mA IP and IgG samples, 6-h 6mA IP and IgG samples from *Drosophila* wild-type embryos, respectively. Adaptor sequences were trimmed, and the reads were then mapped to the *Drosophila* reference genome (UCSC version dm6, BDGP Release 6, unlocalized scaffolds excluded). Bowtie2 software[34] was used with parameters "--no-mixed --no-discordant --non-deterministic --very-sensitive-local" to align the paired-end reads. The callpeak module in MACS2[35] was used to identify enrichment peaks with parameters "-f BAMPE –B" and a default q value (FDR) cutoff of 0.05, with IgG as a control. In total, 17,528, 4,363, 2,447 6mA peaks were identified at the 0.75-, 3-, 6-h wild-type embryos, respectively. To compare 6mA signals from different samples, a fold enrichment method was used to normalize the signals, and the MACS2 bdgcmp module was used to generate the fold enrichment score track with parameters "-m FE –p 2" by comparing wild type and IgG signal tracks in bedGraph.

**Protein–DNA interaction and high-throughput sequencing**. Analysis of the high-throughput sequencing data for Jumu protein–DNA interaction was performed according to the method used to analyze the 6mA-DNA-IP-Seq data.

**Demonstration of the signal of 6mA and Jumu-binding**. Demonstration of the average signal profile of 6mA and Jumu-binding was performed using the SitePro tool from the Cis-regulatory Element Annotation System (CEAS)[36]. The signal distribution on the genome was visualized using the Integrative Genomics Viewer (IGV) software[37].

**Peak annotation**. The genomic distribution of 6mA peaks was annotated by HOMER[38]. We noted that while about 80% of 6mA peaks were less than 400 bp in length, ~3.2% of 0.75-h peaks and 1.4% of 3-h peaks had a width longer than 1 kb (Supplementary Figure 2a). Peaks were considered as gene-associated if the peak center was located between 1 kb upstream from the TSS (Transcription Start Site) and 1 kb downstream from the TTS (Transcription Terminal Site) of a gene. Of note, one feature of the *Drosophila* genome is that intergenic and intronic regions contain numerous simple repeat and transposons, as annotated by the dm6 version of *Drosophila* genome.

**Motif analysis**. Two motif prediction methods, SeqPos in Cistrome[39] and MEME-ChIP[40], were used to obtain enriched motifs on 6mA peak regions. Only sequences located on the central 200 bp from peak summits were considered for motif analysis. We used the top 5,000 6mA peaks from 0.75-h stage to predict the most enriched motifs by SeqPos and obtained the top 10 enriched motifs with the zscore < −15 and the highest hits. Then, total 0.75-h 6mA peaks were used to predict the most enriched motifs by MEME-ChIP, and obtained the top 10 candidates with $E$ value < 0.05 and the highest hits. Motif hits were detected using Find Individual Motif Occurrences (FIMO)[41] with a cutoff p value of 1E-4.

**RNA-seq data analysis**. We performed RNA-seq for wild-type embryo samples at the 0.75-, 3-, and 6-h developmental stages with 125 bp paired-end high-throughput sequencing by Illumina HiSeq 2500 V4 platform and RNA-seq for 3-h wild type, maternal *jumu* mutant, maternal and zygotic *jumu* mutant embryo samples with 150 bp paired-end high-throughput sequencing by NovaSeq platform. The FastQC software (http://www.bioinformatics.babraham.ac.uk/projects/fastqc/) was used to make sure that the sequencing data were high quality. Then, all the sequencing reads that aligned to ribosomal RNA (rRNA) sequences were removed by Bowtie2[34]. The remaining reads were mapped to the December 2014 assembly of the *D. melanogaster* genome (UCSC version dm6, BDGP Release 6, unlocalized scaffolds excluded) using STAR[42] with the default parameters. Differential expression analysis between the different developmental stages was carried out by cuffdiff from the cufflinks package[43]. FPKM (fragments per kilobase of transcript per million fragments mapped) was used to represent gene expression levels. K-means clustering of the differentially expressed genes in the three developmental stages was performed using Cluster 3.0 (clustering method: K-means clustering, Distance metric: Euclidean distance)[44].

To define genes significantly regulated by the maternal Jumu (GSRJ), we used $jumu^{(M-,Z+)}$ and $jumu^{(M-,Z-)}$ mutant embryos RNA-seq data sets at 3-h stage and selected genes with the same regulated direction in $jumu^{(M-,Z+)}$ mutant and $jumu^{(M-,Z-)}$ mutant, compared with wild type, with cuffdiff $q$ value < 0.001.

To define Zelda target genes, we generated the RNA-seq dataset of embryos (3-h stage) with overexpression of maternal Zelda. We integrated this dataset with the Zelda ChIP-seq data[27] published previously to perform the Binding and Expression Target Analysis (BETA)[45] analysis with parameters "--df 0.05 −c 0.05". And finally identified 3,165 Zelda target genes with FDR < 0.001.

**Transposon expression**. *DMAD* knockdown RNA-seq data sets of 6-h embryos were generated by paired-end with 125 bp high-throughput sequencing using Illumina HiSeq 2500 platform V4 by Berry Genomics. The 6-h wild type and *DMAD* knockdown RNA-seq data were aligned to the *Drosophila* genome (UCSC version dm6, BDGP Release 6, unlocalized scaffolds excluded) using STAR[42] with the default parameters and "--outFilterMultimapNmax 100 --winAnchorMultimapNmax 100". The read counts of genes and transposons were calculated by featureCounts[46] and the differential expression analysis was used by DESeq2[47].

**Gene ontology (GO) analysis**. The gene set was uploaded to the GO website (http://geneontology.org/) and PANTHER[48] was used for GO enrichment analysis. The top 20 biological process terms for each group of genes were used to produce a heatmap with the R package pheatmap (https://CRAN.R-project.org/package=pheatmap).

**Proposed regulatory network**. The regulatory network of the Jumu target genes GSRJ and the target genes of Zelda was displayed using Cytoscape[49] v3.6.1.

**Statistical analysis**. $P$ values were calculated by hypergeometric test or Student's $t$ test.

**Reporting Summary**. Further information on research design is available in the Nature Research Reporting Summary linked to this article.

## Data availability

All sequencing data are available at GEO site under accession no. GSE86795. All Public data sets used in this study were downloaded from GEO. The source data underlying Figs. 2a–e, 3b–f, 4a, c–e, 5a, c, e–g, 6a, b and 7a and Supplementary Figures 1a–c, h, 2a–g, 3b–c, 4c, 5a, c, d, f, g, 6a, b, f and 7a–g are provided as a Source Data file.

## Code availability

The custom Perl and R scripts used in this study are available on request to the corresponding authors.

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

## Acknowledgements

The authors thank Dr. Shirley Liu for her critical comments and Yuanyuan Chen and Zhenwei Yang at Core Facility for Protein Research (Institute of Biophysics, CAS) for technical help with Biacore (SPR) experiments. This work was supported by the Ministry of Science and Technology of China (2016YFA0100400, 2018YFC1003300), Strategic Priority Research Program of the Chinese Academy of Sciences (XDB13000000, XDB19000000), Natural Science Foundation of China (31590831, 31671498, 31871294), National Key R&D Program of China (2017YFA0506800, 2016YFC0901700) and CAS Interdisciplinary Innovation Team. Data analysis and computing resource was supported by Center of Big Data Research in Health (http://bigdata.ibp.ac.cn), Institute of Biophysics, Chinese Academy of Sciences.

## Author contributions

S.H., G.Z., Y.T., Q.S., and D.C. designed the experiments; G.Z., Y.G., R.S., Z. Cao, Z. Chen, X.Z., Y.L., X.W., and W.Z. performed experiments; S.H., J.W., J.Y., P.Z., Y.Z., C.H., Y.T., Q.S., and D.C. analyzed data; S.H., G.Z., J.W., Y.T., Q.S., and D.C. wrote the paper. All authors provided intellectual input, vetted and approved the final manuscript.

## Additional information

**Competing interests:** The authors declare no competing interests.

