## [Peer Review File · Nature Communications]

REVIEWERS' COMMENTS:

Reviewer #3

Remarks to the Author:

The authors have partially addressed my concerns but there remains a lack of a clear mechanistic link between Jumu, Zelda and the observed changes in gene expression. While the manuscript claims to have identified a "Jumu-Zelda-ZGA regulatory axis" the data cannot distinguish direct from indirect effects of Zelda, neither do the data permit one to distinguish regulation by Zelda versus by "other factors".

For example, the authors have now separately analyzed Group 1 and Group 2 GSRJ genes relative to Zelda-bound genes. Both Groups are significantly affected: Group 1 is upregulated upon Zelda overexpression while Group 2 is downregulated (response Fig. 5). They claim that the results are "expected". I find this result to be unexpected: since Zelda is a positive regulator of gene expression, why would the effect on Group 2 be down rather than up (or no change)? These data highlight the absence of a mechanistic explanation.

Responses to the reviewer3's comments

Remarks to the Author:

The authors have partially addressed my concerns but there remains a lack of a clear mechanistic link between Jumu, Zelda and the observed changes in gene expression. While the manuscript claims to have identified a "Jumu-Zelda-ZGA regulatory axis" the data cannot distinguish direct from indirect effects of Zelda, neither do the data permit one to distinguish regulation by Zelda versus by "other factors".

Response: First, we would like to thank the reviewer for her/his comments. In this study, we provided compelling genetic evidence showing that Jumu functions as a maternal factor to regulate early embryonic development. Since loss of maternal Jumu leads to up-regulation of Zelda in early embryos, we tried to link the Jumu and Zelda. Our genetic analysis showed that 1) maternal overexpression of Zelda caused embryonic lethal phenotype, which mimics that observed in Jumu maternal mutant embryos; 2) partial loss of Zelda could antagonize the embryonic lethal phenotypes induced by loss of maternal Jumu, suggesting that Zelda is one of critical target genes of Jumu. Moreover, RNA-seq analysis suggested that Jumu-target genes were highly overlapped with Zelda-target genes in early embryos. Because zygotic gene regulation by Zelda is quite complex, we agree with the reviewer's point, we cannot exclude the possibility that Jumu regulates early embryonic development through additional factors. In the final version, we have toned down our conclusion, and avoided to use the "Jumu-Zelda-ZGA regulatory axis".

For example, the authors have now separately analyzed Group 1 and Group 2 GSRJ genes relative to Zelda-bound genes. Both Groups are significantly affected: Group 1 is upregulated upon Zelda overexpression while Group 2 is downregulated (response Fig. 5). They claim that the results are "expected". I find this result to be unexpected: since Zelda is a positive regulator of gene expression, why would the effect on Group 2 be down rather than up (or no change)? These data highlight the absence of a mechanistic explanation.

Response: Previous studies have suggested that loss of maternal Zelda leads to either down-regulation or up-regulation of target genes in *Drosophila* embryos, suggesting that Zelda regulates gene expression in either direct or indirect manner, although Zelda is a positive regulator of gene expression^{1,2}. For the case of indirect targets of Zelda, Zelda could activate a set of miRNAs, which inhibit sets of downstream target genes. Regarding the relationship between Zelda binding and its target gene expression, the situation is also very complicated. It has been shown that 1) not all binding genes were activated by Zelda, only a subset of binding gene was activated, Zelda binding may poise genes for later activation; 2) Zelda's function to activated genes may be mediated, in part, by local histone acetylation and other histone modifications; 3) Zelda-mediated transcriptional activation may be potentiated by the subsequent binding of additional transcription factors³.

References:

1. Fu, S., Nien, C.Y., Liang, H.L. & Rushlow, C. Co-activation of microRNAs by Zelda is essential for early Drosophila development. *Development* **141**, 2108-2118 (2014).
2. Liang, H.L. *et al.* The zinc-finger protein Zelda is a key activator of the early zygotic genome in Drosophila. *Nature* **456**, 400-403 (2008).
3. Schulz, K.N. *et al.* Zelda is differentially required for chromatin accessibility, transcription-factor binding and gene expression in the early Drosophila embryo. *Genome research* (2015).